# Observations of Bromine Monoxide Transport In the Arctic Sustained on Aerosol Particles

Peter K. Peterson[1], Denis Pöhler[2], Holger Sihler[2,3], Johannes Zielcke[2], Stephan General[2], Udo Frieß[2], Ulrich Platt[2,3], William R. Simpson[4], Son V. Nghiem[5], Paul B. Shepson[6], Brian H. Stirm[7], Suresh Dhaniyala[8], Thomas Wagner[3], Dana R. Caulton[9], Jose D. Fuentes[10], and Kerri A. Pratt[1,11]

[1]Department of Chemistry, University of Michigan, Ann Arbor, MI, USA.
[2]Institute of Environmental Physics, University of Heidelberg, Heidelberg, Germany.
[3]Max Planck Institute for Chemistry, Mainz, Germany.
[4]Department of Chemistry and Biochemistry and Geophysical Institute, University of Alaska Fairbanks, Fairbanks, Alaska, USA.
[5]Jet Propulsion Laboratory, California Institute of Technology, Pasadena, CA, USA.
[6]Department of Chemistry, Department of Earth, Planetary, and Atmospheric Sciences, and Purdue Climate Change Research Center, Purdue University, West Lafayette, IN, USA.
[7]School of Aviation and Transportation Technology, Purdue University, West Lafayette, IN, USA.
[8]Department of Mechanical and Aeronautical Engineering, Clarkson University, Potsdam, NY, USA.
[9]Department of Civil and Environmental Engineering, Princeton University, Princeton, NJ, USA.
[10]Department of Meteorology, The Pennsylvania State University, University Park, PA, USA.
[11]Department of Earth and Environmental Sciences, University of Michigan, Ann Arbor, MI, USA.

*Correspondence to:* Kerri Pratt (prattka@umich.edu)

**Abstract.** The return of sunlight in the polar spring leads to production of reactive halogen species from the surface snowpack, significantly altering the chemical composition of the Arctic near-surface atmosphere and the fate of long-range transported pollutants, including mercury. Recent work has shown the initial production of reactive bromine at the Arctic surface snowpack; however, we have limited knowledge of the vertical extent of this chemistry, as well as the lifetime and possible transport of reactive bromine aloft. Here, we present bromine monoxide (BrO) and aerosol particle measurements obtained during the March 2012 BRomine Ozone Mercury EXperiment (BROMEX) near Utqiaġvik (Barrow), AK. The airborne differential optical absorption spectroscopy (DOAS) measurements provided an unprecedented level of spatial resolution, over two orders of magnitude greater than satellite observations and with vertical resolution unable to be achieved by satellite methods, for BrO in the Arctic. This novel method provided quantitative identification of a BrO plume, between 500 m and 1 km aloft, moving at the speed of the air mass. Concurrent aerosol particle measurements suggest this lofted reactive bromine plume was transported and maintained at elevated levels through heterogeneous reactions on co-located supermicron aerosol particles, independently of surface snowpack bromine chemistry. This chemical transport mechanism explains the large spatial extents often observed for reactive bromine chemistry, which impacts atmospheric composition and pollutant fate across the Arctic region, beyond areas of initial snowpack halogen production. The possibility of BrO enhancements disconnected from the surface potentially contributes to sustaining BrO in the free troposphere and must also be considered in the interpretation of satellite BrO column observations, particularly in the context of the rapidly changing Arctic sea ice and snowpack.

# 1 Introduction

With the return of sunlight in the polar spring, production of reactive halogen species radically changes the chemical processes in the Arctic atmospheric boundary layer (Simpson et al., 2007b). Bromine radical chemistry, in particular, is implicated in widespread episodic depletion of boundary layer ozone (Barrie et al., 1988; McConnell et al., 1992; Simpson et al., 2007b), and altered oxidation of atmospheric pollutants, including mercury (Schroeder et al., 1998; Steffen et al., 2008). Satellite observations have given a coarse depiction of the spatial extent of these events (e.g. Richter et al., 1998; Wagner and Platt, 1998; Sihler et al., 2012; Choi et al., 2012), showing that millions of square km are affected by this chemistry. However, many uncertainties remain regarding the initiation of this halogen chemistry and its vertical distribution, as well as the transport and sustained recycling of reactive halogens aloft (Simpson et al., 2015).

Satellite-based observations have shown large plumes (1000s of km) of BrO commonly occurring over sea ice regions in the spring (Wagner and Platt, 1998; Richter et al., 1998). While these plumes could potentially be stratospheric in origin or related to variations in tropopause height (Theys et al., 2009; Salawitch et al., 2010), a variety of tropospheric BrO retrievals (e.g. Theys et al., 2011; Sihler et al., 2012; Koo et al., 2012), which use various techniques to subtract the stratospheric BrO contribution and calculate a tropospheric BrO column, continue to show these large BrO plumes over the Arctic. However, during both the ARCTAS and ARCPAC campaigns, flight measurements showed enhancements in tropospheric BrO above the convective boundary layer (e.g. Salawitch et al., 2010; Choi et al., 2012), potentially sustained on aerosol particles (e.g. Parrella et al., 2012; Schmidt et al., 2016), suggesting some of this BrO column may not be due to boundary layer halogen activation chemistry in sea ice regions.

Ground-based measurements also show a relationship between BrO and sea ice regions (Simpson et al., 2007a); however, the low short-term correlation between time an air mass spends in sea ice regions and the amount of BrO observed suggests while an air mass origin in sea ice regions is important, other environmental factors control the release of reactive bromine (Peterson et al., 2016). Given the short lifetime of BrO in sunlit conditions, heterogeneous reactions are required to explain the amount of reactive bromine observed in the Arctic spring (McConnell et al., 1992; Platt and Hönninger, 2003). While the snowpack provides an abundant surface for these reactions near the ground, both ground-based (Frieß et al., 2011; Peterson et al., 2015) and airborne (McElroy et al., 1999; Prados-Roman et al., 2011; Choi et al., 2012) observations have shown active halogen chemistry also occurring aloft. Concurrent observations of aerosol particle extinction and BrO by Frieß et al. (2011) showed increases in near surface BrO occurring concurrently with high wind speeds and increased aerosol extinction. They interpret this as wind-abraded snow dispersed aloft and providing additional surface area for this chemistry to occur. This process occurred concurrently with snowpack driven halogen chemistry. In contrast, Peterson et al. (2015) observed that, during the spring of 2012, aerosol particle extinction did not depend solely on wind speed, and events occurred with BrO aloft in the absence of enhancements in near-surface aerosol particle extinction. Thus, the mechanism for sustaining such halogen chemistry aloft, independently of the snowpack, remains an open question. While aerosol particles represent a potential surface for these reactions to take place aloft (Fan and Jacob, 1992), there is currently no observational evidence for this phenomenon occurring solely aloft, independently of concurrent snowpack driven halogen activation chemistry.

During the March 2012 BRomine Ozone Mercury EXperiment (BROMEX) near Utqiaġvik (Barrow), Alaska, we investigated the activation (i.e. conversion of Br$^-$ to reactive bromine: e.g. Br atoms, BrO, or HOBr (Fig. 1)) and subsequent transport of reactive bromine with a combination of satellite, airborne, and ground-based measurements (Nghiem et al., 2013). While the spatial extent of this reactive bromine chemistry can be characterized from satellite measurements at a spatial resolution of tens of km, the vertical distribution of BrO in the atmospheric boundary layer cannot be determined via satellite, complicating identification of the source, mechanism of initial activation, subsequent transport scales, and chemical transformation of the reactive bromine. In contrast, ground-based differential optical absorption spectroscopy (DOAS) enables determination of the amount of BrO both near the surface, and in the lowest 2 km of the atmosphere with high temporal resolution (Frieß et al., 2011; Peterson et al., 2015), while airborne DOAS measurements allow determination of spatial distribution of BrO with 100 m vertical and horizontal resolution (General et al., 2014), enabling detailed observation of moving BrO plumes. The combination of these techniques in the current study allowed observations of the evolution of the vertical and horizontal distribution of reactive bromine chemistry with an unprecedented level of detail, enabling identification of the mechanisms driving the transport of reactive bromine aloft.

## 2 Methods

### 2.1 DOAS Measurements

Airborne BrO measurements were made using the Purdue University Airborne Laboratory for Atmospheric Research (ALAR). The 13 March flight took off from the Utqiaġvik airport located at 71.2853° N, 156.7658° W. ALAR carried the Heidelberg Imaging DOAS instrument (HAIDI), allowing for simultaneous measurements in both scanning nadir view and limb view (General et al., 2014). The initial DOAS fitting of the HAIDI spectra and subsequent analysis of the nadir viewing measurements have been described extensively in previous literature (Pratt et al., 2013; General et al., 2014). Limb viewing measurements allowed for retrieval of BrO and aerosol particle extinction profiles (e.g. Prados-Roman et al., 2011) during ALAR ascent and descent. After the initial DOAS fitting, the limb measurements of O$_4$ and BrO were used in a two step optimal estimation procedure to retrieve BrO profiles (Frieß et al., 2011). Averaging kernels, error bars, and further details of these retrievals are given in the supplemental information. Satellite based BrO lower tropospheric vertical column densities (LT-VCDs) were determined using Level 1 GOME-2 data provided by EUMETSAT. The stratospheric subtraction to obtain a LT-VCD was completed using previously published techniques (Sihler et al., 2012).

Ground based MAX-DOAS measurements were conducted at three sites during BROMEX. The instrument at Utqiaġvik was placed on top of the Barrow Arctic Research Consortium building, located at 71.325° N, 156.668° W with a viewing azimuth of 27° east of true north. Additional MAX-DOAS instruments were deployed at two sea ice locations as part of the Icelander platform, an autonomous platform designed to be rapidly deployed via helicopter and conduct chemical and meteorological measurements at remote locations (Peterson et al., 2015). The first Icelander (IL1) was deployed east of Utqiaġvik at 71.355° N, 155.668° W with a view azimuth of 2° east of true north, while the second Icelander (IL2) was deployed to the west at 71.274° N, 157.395° W with a view azimuth of 12° east of true north.

These MAX-DOAS data were analyzed using previously published techniques (Frieß et al., 2011; Peterson et al., 2015). Briefly, solar spectra at telescope elevation angles of 1, 2, 5, 10, and 20° were analyzed using temporally located zenith spectra as a reference for DOAS fitting to retrieve differential slant columns (dSCDs) of BrO and $O_4$. These dSCDs were used to retrieve mole fractions of BrO from the surface up to 2 km with optimal estimation assuming an a priori BrO profile with 10 pmol mol$^{-1}$ at the surface exponentially decaying with a scale height of 250 m. The retrieved profiles represent 20 pieces of information; however, the typical degrees of freedom for signal in our retrievals ranges between 1 and 2 independent pieces of information. Thus, rather than using the retrieved profile, we report a VCD in the lowest 200 m, and if visibility permits, a LT-VCD for 0-2 km (Peterson et al., 2015).

## 2.2 Chemical Ionization Mass Spectrometry

Measurements of $Br_2$, BrO, and HOBr were conducted using chemical ionization mass spectrometry (CIMS) (Peterson et al., 2015) at a site (71.275° N, 156.6405° W) ~5 km inland and southeast of Utqiaġvik, AK. Using $I(H_2O)_n^-$ reagent ions, $Br_2$ was monitored as $I^{79}Br^{81}Br^-$ (287 amu) and $I^{81}Br^{81}Br^-$ (289 amu), BrO as $I^{79}BrO^-$ (222 amu) and $I^{81}BrO^-$ (224 amu), and HOBr as $IH^{79}BrO^-$ (223 amu) and $IH^{81}BrO^-$ (225 amu). For 11-13 March 2012, the limit of detection for 1 h averaging is estimated at 0.4 pmol mol$^{-1}$ for $Br_2$ , BrO, and HOBr, with the associated uncertainties calculated to be (14% +0.4 pmol mol$^{-1}$), (63% +0.4 pmol mol$^{-1}$), and (50% +0.4 pmol mol$^{-1}$), respectively. Additional sampling details are provided in the Supplementary Information.

## 2.3 Aerosol Particle Measurements

Size resolved aerosol particle number concentrations were measured with six second temporal resolution using a Grimm 1.109 optical particle counter (Grimm Aerosol Technik GmbH), which measured particles from 0.25 to 32 $\mu$m in 31 size bins. A size resolved model of inlet transmission was used to correct the measured number concentrations for expected aircraft inlet losses as a function of particle diameter. Details of this model are given in the supplement. After correcting for inlet efficiency, only particles up to 4 $\mu$m (optical diameter) were considered in further analysis. Surface area concentrations were estimated assuming all measured particles were spherical with a diameter equal to the midpoint of the size bin.

## 2.4 Envisat ASAR Sea Ice Images

Satellite radar images were acquired by the Advanced Synthetic Aperture Radar (ASAR) aboard the European Space Agency (ESA)'s Environmental Satellite (Envisat) to support BROMEX. For the period of analysis in this paper, an Envisat ASAR scene was obtained on 11 March 2012 over the BROMEX study domain including the Chukchi Sea and the Beaufort Sea (Figure 2). This ASAR scene was collected in the Wide Swath Medium-resolution mode (ASA_WSM_1P) with the horizontal polarization. ASA_WSM_1P has a geometric resolution of approximately 150 m by 150 m with a pixel spacing of 75 m by 75 m over a swath width of 400 km along a stripe of 4000 km in the maximum extent (Agency, 2007). In the data processing, the precise orbits were used before calibration and terrain correction to achieve accurate backscatter with an excellent geolocation.

The ASAR data allowed an excellent observation of sea ice with the all-weather capability regardless of solar lighting and cloud cover conditions.

## 2.5 Supporting Ozone and Meteorological Measurements

Ozone mole fractions and meteorological data were obtained from the NOAA Global Monitoring Division site near Utqiaġvik, Alaska. Twice daily meteorological soundings (03:00 and 15:00 AKST) were conducted at the Utqiaġvik Airport. The 72 h backward airmass trajectories were calculated using the NOAA Hybrid Single Particle Lagrangian Integrated Trajectory (HYSPLIT) model (Stein et al., 2015). These trajectories were calculated in an ensemble configuration using an arrival height of 750 m agl and one degree GDAS meteorology. Meteorological measurements aboard ALAR were obtained using a "Best Air Turbulence" (BAT) probe (Garman et al., 2006). Airborne $O_3$ measurements were not available during this flight.

## 3 Results and Discussion

Here we show observations of an intense synoptic scale halogen activation event observed at Utqiaġvik, Alaska on 11-13 March. We describe the details of this event chronologically and show the process by which snowpack $Br_2$ production and evolving boundary layer stability later led to the decoupled recycling of reactive bromine on aerosol particles aloft (Fig. 1).

### 3.1 Initial Snowpack-Driven Bromine Activation

During BROMEX, GOME-2 satellite observations of BrO on 11 March 2012 showed an intense synoptic scale halogen activation event over the northern coast of Alaska and Canada, including Utqiaġvik, Alaska (Fig. 3a). Ground-based BrO measurements at both sea ice and tundra sites near Utqiaġvik show this event initiated near the snowpack (Figs. 3b, 4), with 30-40% of the observed BrO column in the lowest 200 m. At Utqiaġvik, winds were out of the northeast with speeds below 6 m s$^{-1}$ (Fig. 5c) which likely enabled movement of air from the snowpack to the overlying atmosphere through wind pumping (Peterson et al., 2015), but were too low to induce blowing snow (Jones et al., 2009; Yang et al., 2010). Further evidence of snowpack sourced reactive bromine is provided by observed enhanced near-surface (1 m) mole fractions of $Br_2$, BrO, and HOBr (Fig. 3). Together, these findings indicate that this event originated at the snow surface, which is consistent with a snowpack source of $Br_2$ (Pratt et al., 2013).

At 17:00 AKST on 11 March, near surface ozone mole fractions sharply decreased at Utqiaġvik from 20 to 8 nmol mol$^{-1}$ over a one hour time period (Fig. 5b). This rapid ozone loss cannot be explained solely by bromine chemistry occurring at Utqiaġvik (Hausmann and Platt, 1994) and likely indicated the advection of a boundary layer ozone depletion event (ODE) into the study area. By 20:00 AKST ozone levels had fallen to near-zero levels (Fig. 5b), which has been shown to inhibit BrO production (Helmig et al., 2012), and is reflected in the BrO observations during this study (Fig. 3b). Inhibition of BrO formation also hindered the formation of HOBr, as shown in Fig. 3b, halting the heterogeneous recycling (Fig. 1) needed to sustain high levels of reactive bromine in the Arctic boundary layer (Fan and Jacob, 1992). The lack of recycling leads to the observed decrease in $Br_2$ production (Pratt et al., 2013), effectively ending snowpack production of reactive bromine (Fig.

3b). This reduction of reactive bromine near the surface is also observed via decreasing BrO column densities observed from GOME-2 on 12 and 13 March (Fig. 3a). Continuous determination of the vertical distribution of BrO by concurrent ground-based DOAS measurements showed the reduction in reactive bromine near the surface was accompanied by movement of BrO aloft on 12 March, transitioning to BrO being observed exclusively aloft on 13 March (Figs. 3b, 4).

## 3.2 Movement of BrO Aloft

Given that the most probable initial source of the observed reactive bromine was the surface snowpack, as discussed above, subsequent observations of BrO aloft raise the question of how the BrO was lofted to $\sim$750 m where the plume was observed. Previously proposed mechanisms for enhancing the vertical extent of BrO events include blowing snow events, where high wind speeds lead to activation of bromine from dispersed snow aloft (Yang et al., 2010; Jones et al., 2009; Frieß et al., 2011), wind pumping events, where increased ventilation of the snowpack in a turbulent atmosphere potentially allows the propagation of BrO aloft (Peterson et al., 2015; Toyota et al., 2014), uplifting of bromine enriched air masses due to synoptic scale meteorological conditions (Frieß et al., 2004; Begoin et al., 2010), and the opening of sea ice leads which are associated with enhanced vertical mixing (Wagner et al., 2007; Moore et al., 2014) and have been proposed to lead to increases in BrO aloft (McElroy et al., 1999).

Backward air mass trajectories (Fig. 6) also suggest the observed lofted bromine originated near the snowpack in sea ice regions as part of the large scale activation event observed on 11 March. These trajectories, along with NCEP reanalysis data (Figs. 6, S4) from 11 March are also consistent with the BrO plume being uplifted from the surface prior to arriving over Utqiaġvik on 13 March. Sea ice imagery (Fig. 2) shows no indication of open sea ice leads near or upwind of Utqiaġvik, ruling out local sea ice dynamics as an explanation for the presence of BrO aloft observed in this study. The vertical distribution of BrO is closely tied to atmospheric stability, with shallow layers associated with thermal inversions, where most of the BrO is in the lowest 200 m, compared to well mixed atmospheres, which allow for propagation of BrO aloft leading to a lower fraction of the observed BrO residing near the surface snowpack (Peterson et al., 2015). Despite the absence of leads, daily meteorological soundings (Fig. 5a) from 11 and 12 March (15:00 AKST) at the Utqiaġvik airport show increased atmospheric mixing allowing BrO to propagate aloft prior to the day of the flight, as seen with the ground-based MAX-DOAS measurements (Fig. 4). On 13 March, the potential temperature profile at flight takeoff showed inhibited atmospheric mixing due to a surface based inversion (Fig. 8), decoupling the chemistry occurring in the plume aloft from the snowpack.

## 3.3 Observed Transport of Reactive Bromine Aloft

On 13 March, individual satellite overpasses (Fig. 7a) showed a troposheric BrO plume linked to the event described above moving to the west over Utqiaġvik. Concurrently, airborne measurements probed both the vertical profile and horizontal distribution of this BrO plume from 10:53 to 13:33 AKST. Figure 9 shows when the aircraft took off from Utqiaġvik, BrO mole fractions were near-zero at the surface increasing to 15-20 pmol mol$^{-1}$ between 500 m and 1 km aloft, indicating that the BrO observed by the satellite during the flight corresponded to a lofted reactive bromine plume. Ground-based MAX-DOAS measurements at Utqiaġvik and two sea ice sites in a linear array spanning $\sim$ 60 km also confirmed the observed BrO remained

aloft as the plume moved westward over the course of the flight (Fig. 7c). Similarly, near-surface measurements of $Br_2$, HOBr, and BrO (Fig. 3b) confirmed that little active halogen chemistry was occurring above the snowpack surface on 13 March, and that any BrO observed during the flight time period was solely due to bromine activation chemistry occurring aloft.

Airborne transects (Fig. 7) enabled detailed characterization of the lofted BrO plume as it moved west. The largest observed BrO lower tropospheric vertical column density (LT-VCD, within the lowest $\sim$2 km) during each airborne transect ranged from 2.8 to $3.3\times10^{13}$ molecules cm$^{-2}$ (Fig. 7b). A linear fit of the plume peak position versus flight time (Fig 7b) for the six plume transects (R=0.97) showed the BrO plume was moving westward with a velocity of 7 m s$^{-1}$, consistent with measured wind speeds of 6-10.5 m s$^{-1}$ to the west at altitudes of 500 m to 1 km during initial flight ascent (Fig. 8). The plume moving with the wind indicates these observations reflect transport of a reactive bromine feature, rather than the plane flying over areas with variable amounts of recycling or primary bromine emission from the snowpack. The vertical profile of BrO obtained just prior to the end of the flight (Fig. 9) showed that the BrO plume was no longer present above Utqiaġvik, consistent with ground-based and satellite observations (Figs. 7a,c).

## 3.4 Role of Aerosol Particles

During plume transport, the peak BrO LT-VCD observed during each transect did not decay during the flight, indicating sustained BrO levels of $\sim$ 20 pmol mol$^{-1}$ (Fig. 9a). Examination of aerosol particle extinction vertical profiles shows the beginning of the flight was marked by a lofted aerosol layer at the altitude of the BrO plume (Fig. 9b) over Utqiaġvik. Like the BrO plume, this lofted aerosol layer was not observed over Utqiaġvik at the end of the flight. The enhanced aerosol particle extinction aloft is consistent with observed enhancements in supermicron ($>$ 1 $\mu$m) aerosol surface area concentrations at 800-900 m (Fig. 9c). During the two transects of the plume conducted in the altitude range where the plume was observed, in situ aerosol particle measurements within the plume showed peaks in supermicron aerosol particle surface area concentrations occurring concurrently with peaks in BrO. For the first transect, both measurements peaked at a longitude of 157.10°W. For the second transect, the BrO peaked at a longitude of 157.31°W, with supermicron aerosol particle surface area peaking at a longitude of 157.36°W, showing the aerosol particle plume and BrO plume were moving together with the wind.

Over the course of this flight, submicron (0.25-1.0 $\mu$m) aerosol particle number concentrations had a median value of 26.5 cm$^{-3}$ with a standard deviation of 16.8 cm$^{-3}$. At altitudes where the plume was observed, the variations of submicron aerosol particle number concentrations within and outside the plume were not significantly different at the 95% confidence level (Smirnov, 1939). Supermicron ($>$1.0 $\mu$m) aerosol particle number concentrations had a median value of 1.2 cm$^{-3}$ with a standard deviation of 6.3 cm$^{-3}$. In contrast to the submicron aerosol, the variability of supermicron aerosol particle number concentrations within and outside the plume were significantly different at the 95% confidence level (Smirnov, 1939), with increasing supermicron aerosol surface area observed with increasing BrO LT-VCDs (Fig. 10). This finding indicates it is unlikely that variations in submicron aerosol particles were a primary controlling factor in the heterogeneous recycling of reactive bromine on particles aloft. However, within the BrO plume, it is likely bromide propagated, through multiphase reactions and subsequent gas-particle partitioning, from supermicron aerosol particles to submicron aerosol particles (Hara

et al., 2002) such that these particles may also have provided additional bromide-enriched surface area for heterogeneous recycling.

Previous studies have shown the majority of supermicron particles in the springtime Arctic boundary layer are sea salt aerosol particles (Brock et al., 2011), which have the potential to be a source of reactive bromine (Fan and Jacob, 1992; Hara et al., 2002). Calculations of the reaction rate of HOBr with bromide based on the measured aerosol particle surface area concentrations and detailed in the supplemental information, suggest the observed surface areas are likely sufficient to enable heterogeneous recycling of reactive bromine aloft. Recent literature (e.g. Yang et al., 2010; Jones et al., 2009) has suggested that wind dispersed saline snow during high wind events could be an important source of bromide-containing aerosol particles. However, Peterson et al. (2015) showed that near surface aerosol particle extinction is not dependent on near surface wind speed at Utqiaġvik and that episodes of enhanced particle extinction and BrO occur most often at lower wind speeds than typically cited as needed for blowing snow (Jones et al., 2009). Recent work by May et al. (2016) showed enhancements in supermicron sea salt aerosol particle number concentrations at Utqiaġvik when wind speeds were greater than 4 m s$^{-1}$ in the presence of open water or sea ice leads; these enhancements were not observed for closed ice conditions. While sea ice imagery (Fig. 2) shows no local lead activity, the lack of available sea ice imagery closer to the origin of the lofted layer (Fig.6) means we can not rule out the possibility of open leads and associated sea salt aerosol production where the observed lofted plume initiated. Given these limitations, we are unable to distinguish between a blowing snow source from consolidated sea ice and sea salt aerosol production from open leads as the source of the sea salt aerosol particles sustaining the BrO plume.

These coincident observations of an aerosol layer and BrO recycling aloft suggest heterogeneous reactions on these aerosol particles played a key role in sustaining the observed enhanced levels of BrO aloft. Although this mechanism was previously proposed (McConnell et al., 1992; Fan and Jacob, 1992), our measurements constitute the first direct observation of this process occurring aloft, decoupled from the snowpack. Further ground-based measurements over the entirety of the BROMEX campaign have confirmed that the presence of aerosol particles aloft is a necessary but not sufficient condition to observe BrO aloft (Simpson et al., 2017).

## 4 Conclusions

The Arctic is currently undergoing rapid changes, including loss of perennial sea ice (Maslanik et al., 2011) and decreasing snow depth in sea ice regions (Blanchard-Wrigglesworth et al., 2015), which impact atmospheric composition through multiphase halogen chemistry (Simpson et al., 2007b). During BROMEX, the horizontal and vertical distribution of reactive bromine, in the form of BrO, was quantified by airborne DOAS at unprecedented spatial resolution, over two orders of magnitude higher than satellite-based measurements. While a pan-Arctic scale view of BrO can be obtained from satellites, this three-dimensional view of the BrO spatial extent, obtained for the first time in the Arctic, combined with higher temporal resolution compared to satellites, provided a process-level view of bromine activation. As summarized in Fig. 1, a synoptic scale BrO event was observed together with snowpack bromine activation, releasing reactive bromine into the Arctic boundary layer. Evolving boundary layer stability allowed the reactive bromine to propagate aloft where it underwent sustained transport

enabled by recycling on aerosol particles independently of snowpack driven halogen activation. The observation of aerosol recycling of reactive bromine enabling the transport of a lofted BrO plume provides a mechanism by which active halogen chemistry alters the fate and transport of atmospheric pollutants far beyond the immediate area where snowpack sourced halogen activation occurs. This mechanism also provides a means of sustaining bromine chemistry in the free troposphere, potentially explaining previous observations of BrO in the Arctic free troposphere (e.g. Salawitch et al., 2010; Choi et al., 2012). This chemical transport mechanism is likely prevalent throughout the springtime Arctic, and the identification of this phenomenon improves our ability to evaluate changes in atmospheric composition related to rapidly changing Arctic sea ice.

## 5 Data availability

Data from the BROMEX field campaign are accessible at: https://nex.nasa.gov/nex/projects/1388/ and/or by contacting the corresponding author.

*Author contributions.* P.K.P. and K.A.P. led data integration, analysis, and interpretation. P.K.P. prepared the initial version of the manuscript and figures. P.K.P. and W.R.S. conducted the ground-based BrO measurements with assistance from U.F., J.Z., and U.P. P.B.S., D.P., S.G., J.Z., K.A.P. and B.H.S. conducted the aircraft-based measurements. H.S. and T.W. provided the satellite based BrO measurements. S.V.N. provided the sea ice measurements. K.A.P and P.B.S. conducted the measurements using chemical ionization mass spectrometry. K.A.P., P.K.P., S.D., and J.D.F. contributed to the aerosol data collection and analysis. D.R.C. contributed to the analysis of aircraft based meteorological measurements. S.V.N. led the 2012 field campaign along with P.B.S. and W.R.S. All authors reviewed and commented on the paper.

*Competing interests.* The authors declare that they have no competing financial interests.

*Acknowledgements.* Financial support for this work was provided by the National Aeronautics and Space Administration (NASA) Earth Science Research Program (NNX14AP44G). Funding for the airborne measurements was provided by NASA Cryospheric Sciences Program (CSP) as a part of the NASA Interdisciplinary Research on Arctic Sea Ice and Tropospheric Chemical Change (09-IDS09-31). The development and construction of the HAIDI instrument was funded by the Deutsche Forschungsgemeinschaft (DFG) within the Priority Program (SPP) No. 1294 "HALO" (DFG PF-384 7/1 and 7/2). The research at the University of Alaska Fairbanks (UAF) was supported by the Department of Chemistry and Biochemistry, NASA CSP, and the National Science Foundation (ARC-1023118). The research at the Jet Propulsion Laboratory, California Institute of Technology, was supported by the NASA CSP and the NASA Atmospheric Composition Program. During the BROMEX campaign, K.A.P. was supported by a National Science Foundation Postdoctoral Fellowship in Polar Regions Research. The authors acknowledge the NOAA Air Resources Laboratory (ARL) for the provision of the HYSPLIT transport and dispersion model used in this publication. Alexei Rozanov (IUP Bremen) is thanked for providing the SCIATRAN radiative transfer code. Steve Walsh (UAF) is thanked for assistance with construction and deployment of the Icelander platforms. Kyle Custard (Purdue University),

David Tanner (Georgia Tech), and Greg Huey (Georgia Tech) are thanked for assistance with CIMS measurements. National Centers for Environmental Prediction (NCEP) reanalysis data were provided by the Physical Sciences Division, Earth System Research Laboratory, NOAA (http://www.esrl.noaa.gov/psd/). GOME-2 level 1 data were provided by EUMETSAT. We thank Pablo Clemente-Colón and Christopher Jackson of the U.S. National Ice Center for their help in accessing the Envisat ASAR data, and Lisa Nguyen at the Jet Propulsion Laboratory for the SAR data processing and geolocation.

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

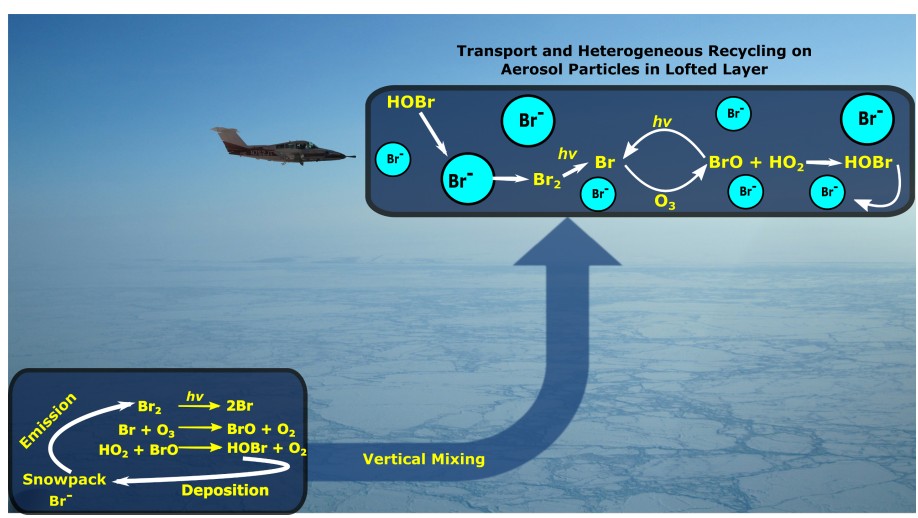

**Figure 1.** Overview of processes observed in this study. Bromide (Br$^-$) is converted to reactive bromine (e.g. Br$_2$, BrO, HOBr) through heterogeneous reactions occurring in the surface snowpack leading to the release of Br$_2$ into the overlying atmosphere. This Br$_2$ undergoes photolysis, creating bromine atoms and leading to ozone depletion. As increased mixing is observed, the bromine enriched air mass moves aloft. On the day of the flight, we observed sustained BrO levels in a lofted layer concurrently with aerosol particles, indicating heterogeneous recycling of reactive bromine aloft, decoupled from the initial snowpack activation processes.

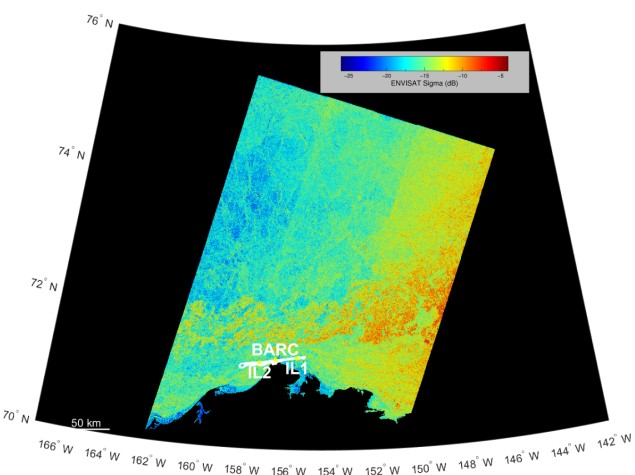

**Figure 2.** Sea ice imagery from Envisat Advanced Synthetic Aperture Radar (ASAR) backscatter data (Sigma in dB). Flight track and measurement locations; Barrow Arctic Research Consortium (BARC) building, Icelander 1 (IL1), and Icelander 2 (IL2) are shown. Multi-year ice is shown in darker red and first-year ice in orange to green and blue colors.

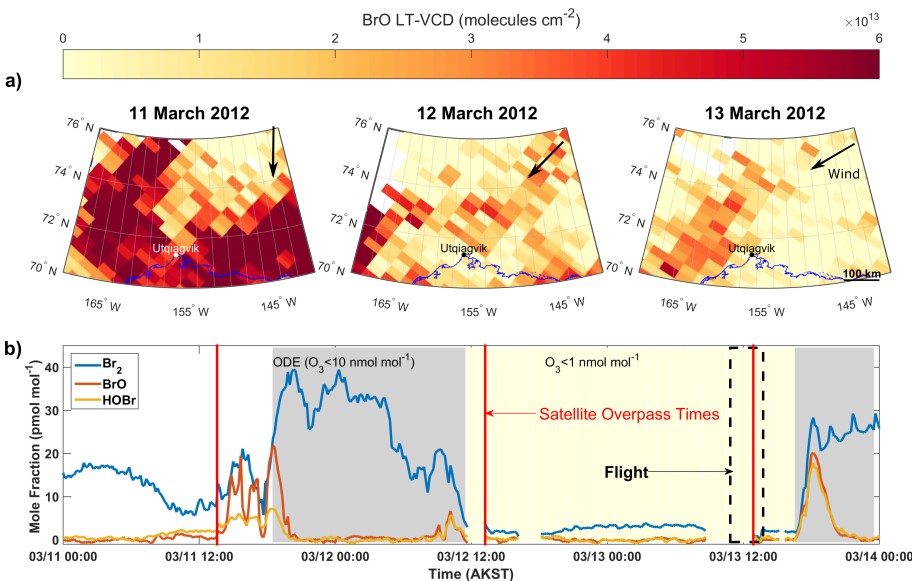

**Figure 3.** Summary of a) satellite based BrO observations over northern Alaska and b) ground-based reactive bromine measurements at the Utqiaġvik tundra site. a) BrO lower tropospheric vertical column densities (LT-VCD) from the GOME-2 satellite are shown from 11-13 March 2012, with the location of Utqiaġvik, Alaska denoted. The top portion of the map is the Arctic Ocean, while the bottom portion is snow covered tundra. The wind direction at Utqiaġvik is shown with black arrows. b) Gas phase bromine species (Br$_2$, BrO, HOBr) mole fractions measured with chemical ionization mass spectrometry are shown for 11-13 March 2012. Data gaps represent time periods of separate experiments (Pratt et al., 2013). Times when an ODE (O$_3$<10 nmol mol$^{-1}$) was observed at Utqiaġvik are shaded in gray; times when ozone was titrated (O$_3$<1 nmol mol$^{-1}$) to the point of inhibiting BrO formation are shaded in yellow. The flight time is outlined with a black box, and the times of satellite overpasses are denoted with red lines.

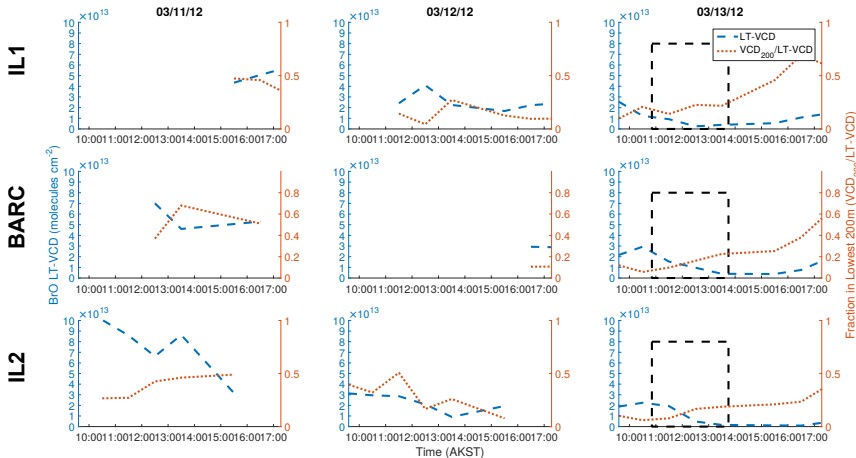

**Figure 4.** Summary of ground based MAX-DOAS BrO observations near Barrow, all times are local. Observations are from IL1 (top row), the BARC building (middle), and IL2 (bottom row). The blue lines show the LT-VCD measured from the ground, and the red lines show the fraction in the lowest 200 m. LT-VCDs are not shown when the degrees of freedom aloft is below 0.7, indicating reduced visibility.

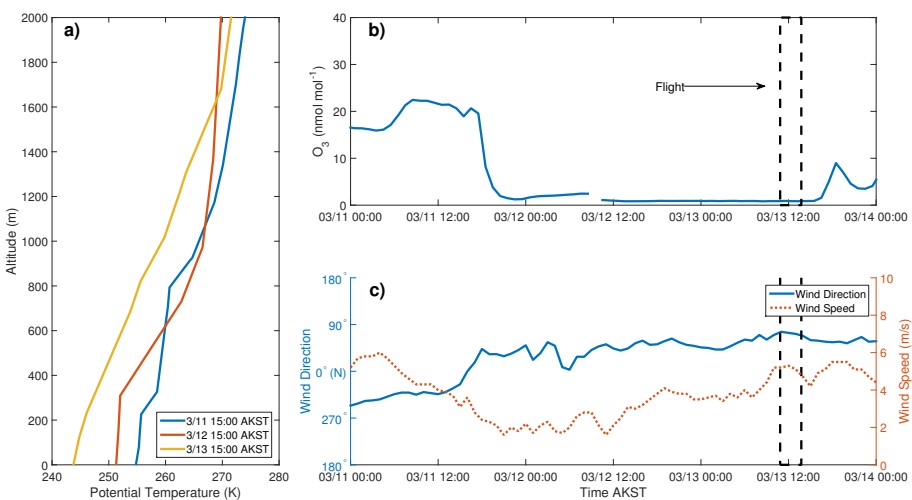

**Figure 5.** Meteorological and in-situ ozone measurements from Utqiaġvik. a) Daytime soundings from the Barrow Airport. b) Near-surface ozone mole fractions from the NOAA/GMD Barrow Observatory. c) Wind speeds and directions from the NOAA/GMD Barrow Observatory. On panels b and c, rectangles indicate when the flight took place.

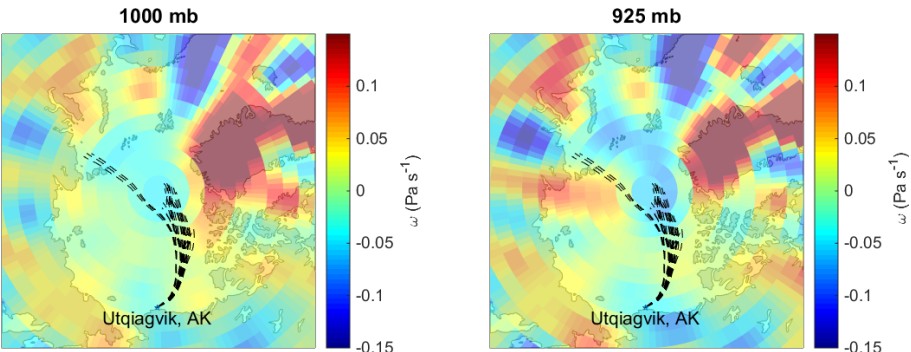

**Figure 6.** NOAA generated HYSPLIT backward air mass trajectories for the plume overlaid on daily averaged NCEP reanalysis data for 11 March 2012. All panels show an ensemble of twenty seven 72 h trajectories arriving at Utqiaġvik on 13 March 2012 at noon local time. The central trajectory had an arrival height of 750 m. The ensemble is generated by offsetting the modeled meteorology used to calculate the trajectory by 1° horizontally and ∼250 m vertically prior to running HYSPLIT. The blue regions on the map reflect regions where air was moving upward, while red regions reflect regions where air was moving downward.

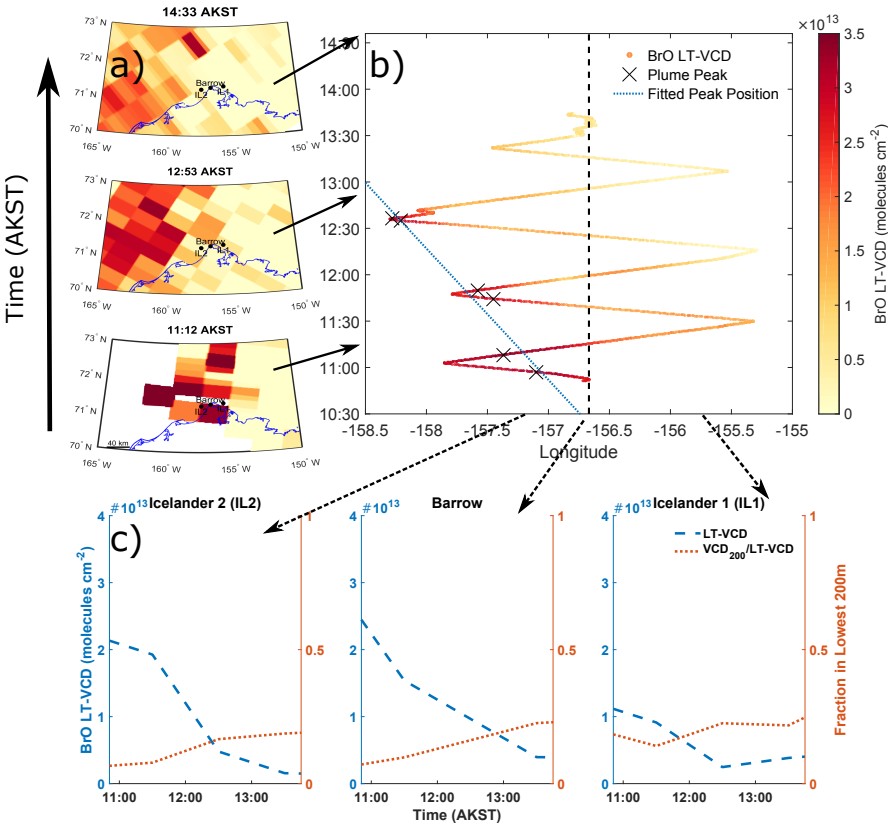

**Figure 7.** Summary of BrO plume motion over Utqiaġvik on 13 March. a) BrO plume motion from east to west as observed by individual overpasses of the GOME-2 satellite as well as the locations of ground-based measurement sites. The blue line denotes the Alaska coastline. b) BrO plume motion as observed by airborne DOAS. The dashed black line indicates the longitude of Utqiaġvik. The black crosses show the observed peak BrO positions for each transect, while the blue line gives the fitted plume peak location as a function of time. c) Ground-based MAX-DOAS measurements at each site during the flight arranged from west to east.

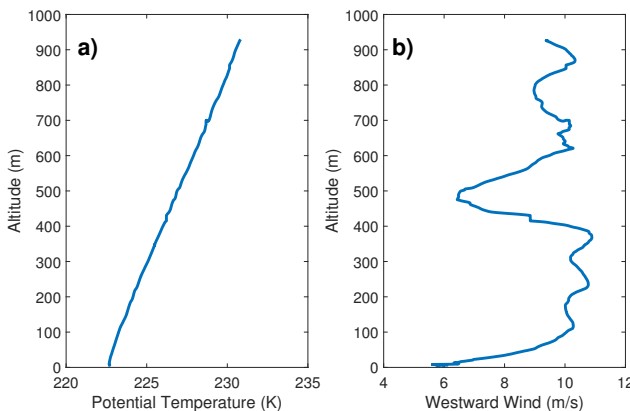

**Figure 8.** Potential temperature (a) and east-west component of wind speed profiles (b) from aircraft ascent on 13 March 2012 at 10:53 AKST.

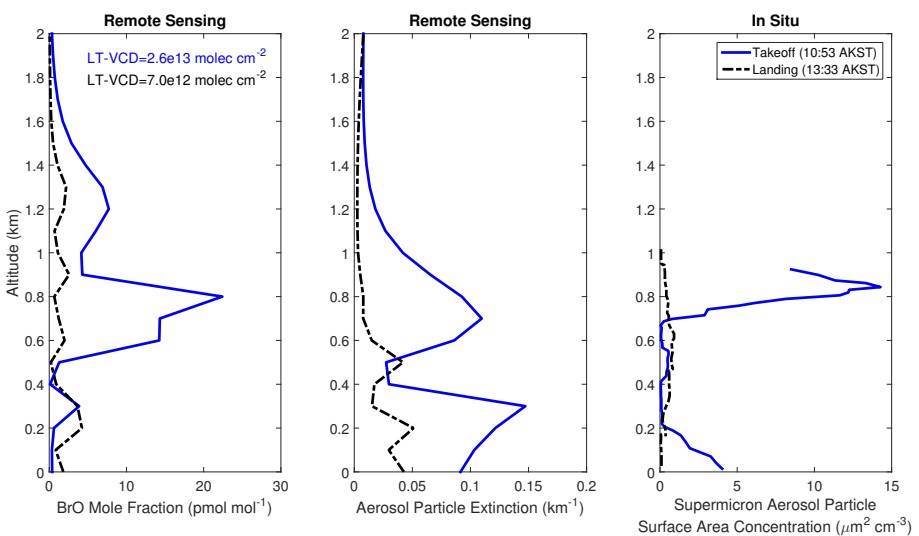

**Figure 9.** Vertical profiles of BrO mole fractions with the corresponding LT-VCD, aerosol particle extinction, and supermicron (1-4 $\mu$m optical diameter) aerosol particle surface area concentrations during takeoff and landing on the 13 March 2012 flight. Error bars on the remote sensing measurements have been omitted for clarity and are shown in the supplemental information.

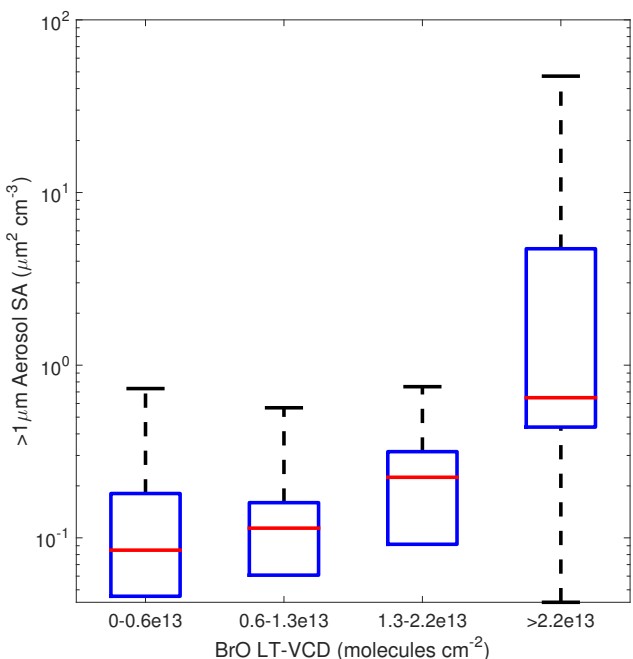

**Figure 10.** Box and whisker plot of supermicron particle surface area concentrations as a function of BrO LT-VCD. The red line represents the median, the blue box outlines the inner quartiles, and the whiskers represent the outer quartiles. In some cases, the difference between the lowest aerosol observation and the 25th percentile is too small to draw a lower whisker. These aerosol observations are restricted to altitudes where the plume was observed on takeoff and binned by dividing the BrO observations into quartiles.

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
