# Peer review of "Observations of Bromine Monoxide Transport In the Arctic Sustained on Aerosol Particles"

_Atmospheric Chemistry and Physics, 2016_

## Referee Comment (RC1) · Anonymous Referee #1 · 20 Mar 2017

This paper presents a detailed analysis of a bromine activation event that took place over Barrow in March 2012. This event was probed by BrO and supporting measurements at the surface (in situ and MAX-DOAS), from aircraft (MAX-DOAS), and from satellite. A spectacular finding is the observation of a BrO plume aloft at 0.5-1 km altitude, implying that BrO can be sustained through reservoir recycling reactions within the atmosphere presumably through aerosols. This is a major new finding not only for understanding elevated BrO in Arctic spring but also for explaining the BrO tropospheric background. This paper certainly deserves publication in ACP.

The paper can be published pretty much as is in my opinion (I do think the abstract needs some tweaking, see below). However, I found it a struggle to go through because of all the complicated plots of near-raw data forcing me through (I thought) unnecessary details. In my view, Figure 11 makes the paper. The authors might consider cutting

back on the figures or simplifying them. This is just a suggestion, however, because the authors might feel that the detail is necessary. Readers like me will be discouraged by the detail and go straight to the abstract, and that's OK.

I do think that the authors can (and should) broaden the impact of their paper by linking their results to background tropospheric BrO. As they know, there is a lot of current interest in understanding the ∼1ppt BrO observed in the troposphere with implications for ozone, OH, and Hg. Heterogeneous recycling in aerosols similar to the springtime Arctic (but at a slower pace) has been proposed to explain the sustained background tropospheric BrO levels but without direct evidence (Parrella et al., ACP 12, 6723-6740, 2012; Schmidt et al., JGR 121, 11819-11835, 2016; sorry to be pushing my own literature). The present paper offer strong support for this heterogeneous recycling. It would be a neat way to connect the Arctic spring chemistry to the global picture. Brief statements in the Abstract, intro, and conclusions is all that would take.

Specific comments: (page, line) 1. Abstract, line 9: "disconnected from the surface" is vague, I would point out that the plume is at 0.5-1 km altitude.

2. Abstract, lines 10-11: I don't think that the authors can claim as fact that the recycling took place on the co-located supermicron aerosol particles. They can claim evidence that it did. It's too bad that the aircraft didn't carry in situ instrumentation that would provide more correlative information including BrO reservoirs and ozone as well as aerosols. Maybe for the next campaign?

3. Abstract, line 12: "increases the spatial extent of bromine chemistry" This is vague. We already know that BrO events extend for great distances horizontally. Maybe the authors mean vertical extent? That would definitely be an appropriate statement.

4. Abstract, line 14: "...must be considered in the interpretation of satellite observations..." Vague, how is that? I didn't see reference to this in the text.

5. Page 7, line 7: give the source of the tropospheric BrO satellite data (a website can

be a useful reference). How was the stratosphere removed?

6. Page 11: a spectacular observation from this paper is that BrO is "killed" by ozone depletion at the surface but apparently not at 0.5-1 km. This led me to search through the paper for any indication of ozone data aloft but I didn't find any. Presumably there were no ozone measurements from the aircraft? That would be worth stating. No ozonesondes? Does the general ozonesonde climatology in the Arctic have information on the vertical structure of ozone depletion in the lowest 2 km?

7. As an example of making figures more user-friendly, it would help if the LT-VCD data and the fraction below 200 m were expressed in units of ppb BrO. More generally, anything you can do to cut back on the number of figures (or panels within figures) or simplify them would (I think) be appreciated by the reader.

---

## Referee Comment (RC2) · Anonymous Referee #2 · 16 Apr 2017

[Summary]

In this study, Peterson et al. analyzed the spatial and temporal evolution of gaseous reactive bromine species (especially BrO) around Utqiagvik (Barrow) measured from a suite of platforms between March 11 and 13 in 2012 during the BROMEX campaign.

Three MAX-DOAS instruments deployed at the ground/sea levels, one on the land (BARC site) and the other two on the sea ice to the west and east of the BARC site, provided the continuous time series of vertical column densities in the lowest 200 m (VCD200) and lower tropospheric vertical column densities (LT-VCDs) for BrO during the daytime. GOME-2 satellite retrievals provided pan-Arctic distributions of BrO LT-VCDs once per day and, because of swath overlaps, a few times per day over the small area of interest in this study around Utqiagvik. Airborne DOAS measurements

on March 13 gave aerial surveys for a few hours along (and beyond) the line of the three MAX-DOAS surface deployments, probing much finer horizontal structures of the BrO LT-VCDs than can be seen from GOME-2 by nadir view during the cruise flight as well as the vertical profiles of BrO and aerosol extinction during the ascent and decent of the aircraft by limb view. The aircraft also measured the size resolved aerosol number concentrations in-situ, complementing the vertical aerosol extinction profiles retrieved by DOAS. Also used for the present analysis are the ground based measurements of $Br_2$, BrO and HOBr by CIMS at a location several kilometers inland from Utqiagvik, surface ozone and wind measurements at the NOAA/GMD site and once-per-day meteorological soundings from Barrow Airport. Supplemented by the ASAR sea ice imagery and backward trajectories, the authors documented, at the "unprecedented" spatial resolution, three-dimensional structures and their time evolutions of air masses enriched in reactive bromine presumably released from the surface snowpack including the synoptic-scale lofting of reactive bromine to the height detached from the surface.

This case study is an important contribution to the field and hence I support its publication subject to minor revisions.

[Specific comments]

1. Section 3.1: I am not too impressed to the presentation order of Figure 4-7. The story telling in Section 3.1 begins with referral to Figures 5 and 7, followed by Figures 7 and 6 and then Figure 4. I felt uneasy while flipping pages back and forth many times until I understood the basic story. I suggest the authors to reorganize either the figure presentation order or the sentence order in Section 3.1 if at all possible.

2. Section 3.2, second paragraph: The description of the atmospheric stability on March 11 and 12 and its link to the mixing of BrO could be more specific. From the vertical gradient of potential temperatures, I can see that the lowest 200 meters of air was well mixed on March 11 and the lowest 300 meters on March 12. On March 11,

however, there was a second layer between 300 and 800 meters where the air appears to have been mixed relatively well. If the authors believe that this second layer carried a significant amount of surface-sourced reactive bromine, they should say so explicitly.

3. Section 3.3: You can make more sense of the vertical profile of BrO mixing ratios during the aircraft takeoff shown in Figure 11 by integrating over 0-2km altitudes and then comparing with the values of BrO LT-VCDs shown in Figure 9.

4. Section 3.4: It is not always clear to me what the authors mean by "heterogeneous recycling", especially when they refer to the Hara et al. (2002) study at the end of the second paragraph. I normally use the term "recycling" when referring to conversion of gaseous halogen species into a more photolabile one, e.g., $HOBr + HBr \rightarrow Br_2 + H_2O$. If you refer to the reaction $HOBr + Br^- + H^+ \rightarrow Br_2 + H_2O$ where the bromide anion is directly provided from sea salt, I would call it a "bromine explosion". I would like the authors to state this difference a little more clearly in Section 3.4. The abundance of the super-micron aerosols measured in-situ at 700-1000 meters aloft (Fig. 11) does seem to imply these particles are relatively "fresh" having been emitted either from open leads or snow-covered surfaces, even though I understand the reservations by the authors as discussed in detail.

[Technical suggestions]

P2, L13 & L16: "e.g" -> "e.g."

P2, L28-29: Add a comma between "that" and "during".

P4, L12: "Sihler et al. (2012)" -> "(Sihler et al., 2012)"

P10, Figure 8 caption: Add the arrival date (March 13th) and time (either in UT or AKST) of the trajectories over Utqiagvik. Also, add the height (750 m) of central trajectory arrival.

P10, L19: "daily soundings" -> "daily meteorological soundings"

P12, Figure 10b (x-axis title): "West Wind" -> "Westward Wind"

P12, L17: "Fig. 11" -> "Fig. 11a"

P12, L17: "show" -> "shows"

P12, L18: "Fig. 11" -> "Fig. 11b"

P12, L21: "Fig. 11" -> "Fig. 11c"

P13, Figure 11: Change the x-axis title "BrO Molar Ratio" -> "BrO Mixing Ratio".

P13, L12: "an initial" -> "a primary"

P13, L14: Remove the comma after "particles".

P15, the top line: "or" -> "and"

---

## Author Response (AR1)

Reviewer Response: acp-2016-1141

**We thank Anonymous Referee #1 for the constructive comments on this manuscript. In our response, we first address the overarching concerns of the reviewer, followed by addressing individual comments. Our response to points that the reviewer raises is denoted with bold text. All line numbers in our responses refer to the attached tracked change manuscript.**

This paper presents a detailed analysis of a bromine activation event that took place over Barrow in March 2012. This event was probed by BrO and supporting measurements at the surface (in situ and MAX-DOAS), from aircraft (MAX-DOAS), and from satellite. A spectacular finding is the observation of a BrO plume aloft at 0.5-1 km altitude, implying that BrO can be sustained through reservoir recycling reactions within the atmosphere presumably through aerosols. This is a major new finding not only for understanding elevated BrO in Arctic spring but also for explaining the BrO tropospheric background. This paper certainly deserves publication in ACP. The paper can be published pretty much as is in my opinion (I do think the abstract needs some tweaking, see below). However, I found it a struggle to go through because of all the complicated plots of near-raw data forcing me through (I thought) unnecessary details. In my view, Figure 11 makes the paper. The authors might consider cutting back on the figures or simplifying them. This is just a suggestion, however, because the authors might feel that the detail is necessary. Readers like me will be discouraged by the detail and go straight to the abstract, and thats OK.

**Based on the reviewer's suggestion, Figs. 2 and 5 have been moved to the supplemental information. In addition, the now Fig. 9 has been modified to show the LT-VCD as well as the BrO profile to facilitate comparisons between figures.**

I do think that the authors can (and should) broaden the impact of their paper by linking their results to background tropospheric BrO. As they know, there is a lot of current interest in understanding the ∼1ppt BrO observed in the troposphere with implications for ozone, OH, and Hg. Heterogeneous recycling in aerosols similar to the springtime Arctic (but at a slower pace) has been proposed to explain the sustained background tropospheric BrO levels but without direct evidence (Parrella et al., ACP 12, 6723- 6740, 2012; Schmidt et al., JGR 121, 11819-11835, 2016; sorry to be pushing my own literature). The present paper offer strong support for this heterogeneous recycling. It would be a neat way to connect the Arctic spring chemistry to the global picture. Brief statements in the Abstract, intro, and conclusions is all that would take.

**This is an excellent point. As suggested, we have added brief statements to the abstract (pg. 1, line 15), intro (pg. 2, line 19) and conclusions (pg. 15, line 5) discussing the implications of this finding for the tropospheric background as**

**suggested by the reviewer.**

Specific comments: (page, line) 1. Abstract, line 9: disconnected from the surface is vague, I would point out that the plume is at 0.5-1 km altitude.

**We changed this sentence to specify the plume altitude.**

Abstract, lines 10-11: I dont think that the authors can claim as fact that the recycling took place on the co-located supermicron aerosol particles. They can claim evidence that it did. Its too bad that the aircraft didnt carry in situ instrumentation that would provide more correlative information including BrO reservoirs and ozone as well as aerosols. Maybe for the next campaign?

**We have modified this sentence to suggest this recycling took place rather than explicitly say this took place. While the plane did carry an ozone monitor, it was unfortunately not functional for this flight. We are intending to pursue more detailed gas and aerosol chemistry measurements in the future.**

Abstract, line 12: increases the spatial extent of bromine chemistry This is vague. We already know that BrO events extend for great distances horizontally. Maybe the authors mean vertical extent? That would definitely be an appropriate statement.

**The mechanism allows for transport far from the source region, which extends the impacts of this halogen chemistry in both the horizontal and vertical dimension. While we already know BrO events extend for large distances horizontally, these results provide evidence for a likely process governing the BrO spatial extents observed by satellite measurements. This is now clarified in the abstract.**

Abstract, line 14: must be considered in the interpretation of satellite observations Vague, how is that? I didnt see reference to this in the text.

**We argue that the existence of plumes disconnected from the surface would alter the interpretation of satellite observations of enhanced LT-VCDs. We now clarify that we are referring to column observations**

Page 7, line 7: give the source of the tropospheric BrO satellite data (a website can be a useful reference). How was the stratosphere removed?

**We have modified the text in the methods (pg. 4, lines 15-16) to clarify the answer to these questions.**

Page 11: a spectacular observation from this paper is that BrO is killed by ozone depletion at the surface but apparently not at 0.5-1 km. This led me to search through the paper for any indication of ozone data aloft but I didnt find any. Presumably there were no ozone measurements from the aircraft? That would be worth stating. No ozonesondes? Does the general ozonesonde climatology in the Arctic have information on the vertical structure of ozone depletion in the lowest 2 km?

**We have added a statement (pg. 6, line 1) that no ozone measurements aloft are available for this flight. Oltmans et al. (2012) showed that ODEs can vary between hundreds of meters to 2km in thickness and the transition between completely depleted air near the surface and non depleted air aloft is typically gradual.**

As an example of making figures more user-friendly, it would help if the LT-VCD data and the fraction below 200 m were expressed in units of ppb BrO.

**We have modified Fig. 9 to include LT-VCD information to facilitate comparison to other figures. However, converting a LT-VCD to a mixing ratio requires one to make assumptions about the vertical distribution of BrO, which we show in this paper is quite variable. Thus, we do not feel it is appropriate to make conversions between LT-VCDs and mole fractions in the absence of detailed vertical profile information (figures other than Fig. 9).**

More generally, anything you can do to cut back on the number of figures (or panels within figures) or simplify them would (I think) be appreciated by the reader.

**The original figures 2 and 5 have been moved to the supplement.**

We thank Anonymous Referee #2 for the constructive comments on this manuscript. Our response to points that the reviewer raises is denoted with **bold text.**

Section 3.1: I am not too impressed to the presentation order of Figure 4-7. The story telling in Section 3.1 begins with referral to Figures 5 and 7, followed by Figures 7 and 6 and then Figure 4. I felt uneasy while flipping pages back and forth many times until I understood the basic story. I suggest the authors to reorganize either the figure presentation order or the sentence order in Section 3.1 if at all possible.

**We have reorganized the presentation order of the figures to match the referrals in the text.**

Section 3.2, second paragraph: The description of the atmospheric stability on March 11 and 12 and its link to the mixing of BrO could be more specific. From the vertical gradient of potential temperatures, I can see that the lowest 200 meters of air was well mixed on March 11 and the lowest 300 meters on March 12. On March 11, however, there was a second layer between 300 and 800 meters where the air appears to have been mixed relatively well. If the authors believe that this second layer carried a significant amount of surface-sourced reactive bromine, they should say so explicitly.

**Ground based measurements (Fig. 5) showed the majority of the BrO was in the lowest 200 m of the atmosphere, thus we do not believe this layer observed on the 11th carried a significant amount of surface-sourced bromine.**

Section 3.3: You can make more sense of the vertical profile of BrO mixing ratios during the aircraft takeoff shown in Figure 11 by integrating over 0-2km altitudes and then comparing with the values of BrO LT-VCDs shown in Figure 9.

**We modified the now Figure 9 to include LT-VCD information.**

Section 3.4: It is not always clear to me what the authors mean by heterogeneous recycling, especially when they refer to the Hara et al. (2002) study at the end of the second paragraph. I normally use the term recycling when referring to conversion of gaseous halogen species into a more photolabile one, e.g., $HOBr + HBr ->Br2 + H2O$. If you refer to the reaction $HOBr + Br- + H+ ->Br2 + H2O$ where the bromide anion is directly provided from sea salt, I would call it a bromine explosion. I would like the authors to state this difference a little more clearly in Section 3.4. The abundance of the super-micron aerosols measured in-situ at 700-1000 meters aloft (Fig. 11) does seem to imply these particles are relatively fresh having been emitted either from open leads or snow-covered surfaces, even though I understand the reservations by the authors as discussed in detail.

**Heterogeneous recycling in this context refers to heterogeneous reactions that regenerate BrO$_x$ from HOBr and HBr that has deposited on particles. We have modified the text (pg. 12, line 21) in this section to clarify this point.**

P2, L13 & L16: e.g ->e.g.

**We made this change.**

P2, L28-29: Add a comma between that and during.

**We made this change.**

P4, L12: Sihler et al. (2012) ->(Sihler et al., 2012)

**We made this change.**

P10, Figure 8 caption: Add the arrival date (March 13th) and time (either in UT or AKST) of the trajectories over Utqiagvik. Also, add the height (750 m) of central trajectory arrival.

**We made this change.**

P10, L19: daily soundings ->daily meteorological soundings

**We made this change.**

P12, Figure 10b (x-axis title): West Wind ->Westward Wind

**We made this change.**

P12, L17: Fig. 11 ->Fig. 11a

**We made this change.**

P12, L17: show ->shows

**We made this change.**

P12, L18: Fig. 11 ->Fig. 11b

**We made this change.**

P12, L21: Fig. 11 ->Fig. 11c

**We made this change.**

P13, Figure 11: Change the x-axis title BrO Molar Ratio ->BrO Mixing Ratio.

**We changed this title to "BrO Mole Fraction" to match the text.**

P13, L12: an initial ->a primary

**We made this change.**

P13, L14: Remove the comma after particles.

**We made this change.**

P15, the top line: or ->and

**We made this change.**

**References**

Oltmans, S. J., Johnson, B. J., and Harris, J. M.: Springtime boundary layer ozone depletion at Barrow, Alaska: Meteorological influence, year-to-year variation, and long-term change, Journal of Geophysical Research, 117, D00R18, doi:10.1029/2011JD016889, URL `http://doi.wiley.com/10.1029/2011JD016889`, 2012.

[revised manuscript text omitted]

---

## Author Response (AR2)

Co-editor Response: acp-2016-1141

In our response, our response to points that the editor raises is denoted with bold text. All line numbers in our responses refer to the final manuscript.

(1) the manuscript Simpson et al (ACP-2017-187) should probably at least be mentioned, especially since it digs further in the question of the aerosol impact of recycling BrOx.

**We added a reference to this work on Page 8, Line 21.**

(2) do you really have to mention always that Utqiagvik is (Barrow)? To me it suffices to mention it the first time it occurs and go from there just using Utqiagvik. By the way, while nitpicking on this, I note that you are not completely consistent anyway (see P8/L15, finally (Barrow) is missing.. ). And to make matters worse, the Simpson et al paper uses (consistently I believe) "Barrow (Utqiagvik)" rather than "Utqiagvik (Barrow)".

**We made this change, electing to always refer to it as Utqiagvik, since that is the current name of the town.**

(3) and while nitpicking: P4/L2: "using a temporally ... spectra", delete the "a", and P5/L8: "arrival Heights", delete the "s"

**We made these changes.**

[revised manuscript text omitted]